# The Phase Space Analysis of Interacting K-Essence Dark Energy Models in Loop Quantum Cosmology

**Bohai Chen** [1,2], **Yabo Wu** [1,*], **Jianan Chi** [1], **Wenzhong Liu** [1]  **and Yiliang Hu** [1]

1    Department of Physics, Liaoning Normal University, Dalian 116029, China
2    School of Liberal Arts and Sciences, North China Institute of Aerospace Engineering, Langfang 065000, China
*    Correspondence: ybwu@lnnu.edu.cn

**Abstract:** The present work deals with two kinds of k-essence dark energy models within the framework of loop quantum cosmology (LQC). The two kinds of k-essence models originates from two forms of Lagrangians, i.e., $\mathcal{L}_1 = F(X)V(\phi)$ and $\mathcal{L}_2 = F(X) - V(\phi)$, where $F(X)$ and $V(\phi)$ stand for the kinetic term and potential of the scalar field $\phi$, respectively. Two models are based on different phase variables settings, and the general form of autonomous dynamical system is deduced for each Lagrangian. Then, the dynamical stabilities of the critical points in each model are analysed in different forms of $F(X)$ and $V(\phi)$. Model I is a 3-dim system with four stable points, and Model II is a 4-dim system but reduced to a 3-dim system using the symmetry analysis, which has five stable points. Moreover, the corresponding cosmological quantities, such as $\Omega_\phi$, $w_\phi$ and $q$, are calculated at each critical point. To compare these with the case of the classical Einstein cosmology (EC), the dynamical evolutionary trajectories in the phase space and evolutionary curves of the cosmological quantities are drawn for both EC and LQC cases, which shows that the loop quantum gravity effects diminish in the late-time universe but are significant in the early time. Further, the effects of interaction $Q = \alpha H \rho_m$ on the evolutions of the universe are discussed. With the loop quantum gravity effects, bouncing universe is achieved in both models for different initial values of $\phi_0$, $\dot{\phi}_0$, $H_0$, $\rho_0$ and coupling parameter $\alpha$, which helps to avoid singularities. However, the interaction has little effect on bounce, although it is important to the stability of some critical points.

**Keywords:** k-essence dark energy; loop quantum cosmology; dynamical stability; interaction



## 1. Introduction

As we know, the astronomical observations on the luminosity–redshift relation of distant Type Ia supernovas [1,2] in the late 90's gave the world a striking surprise that the universe is experiencing an accelerated expansion. This mysterious power is the so-called dark energy (DE), which has the feature of negative pressure and its equation of state (EoS) $w_{DE} = \frac{p_{DE}}{\rho_{DE}} < -1/3$, where $p_{DE}$ is its pressure and $\rho_{DE}$ is the density. It motivates the further observations and theoretical researches. Over the last twenty years, with the enhancement of observational accuracy [3–5], measurements indicate that the current energy density in the universe is composed of 68.3% dark energy, 26.8% dark matter (DM) and 4.9% baryons; meanwhile, the relative constraints on cosmological quantities were developed from different aspects [6–9]. Until now, the ΛCDM model is still a decent explanation to the accelerated expansion, which has a constant equation of state (EoS) $w_\Lambda = -1$ as a candidate of DE in accordance with the observations. However, the ΛCDM model is not so perfect to unify all problems, such as reconciling the cosmological constant problem, the age problem and the coincidence problem [10–14]. Then, phenomenally a vast of scalar field models have been researched to replace the cosmological constant Λ, while generally in those models pressure and energy density are expressed by certain forms of kinetic and potential in terms of a scalar and its derivative to time, such as, quintessence, phantom, k-essence and Dirac-Born-Infeld models, etc. [15–21].

However, all the above models face the singularity problem, such as phantom field DE in classical Einstein cosmology, which leads to the Big Rip singularity for the fluids with $w \ll -1$ [22]. Following the classical Einstein cosmology (EC), based on quantum geometry, loop quantum gravity (LQG) was developed as a background-independent, non-perturbative approach to quantum gravity, and has been applied as loop quantum cosmology (LQC), which adds a modified term $(1 - \frac{\rho}{\rho_{lc}})$ to the classical Friedmann equation [23], as well as some other forms, as in [24,25]. Within the framework of LQC, this provides us with a novel means of researching the big bang singularity and the black hole singularity [26,27]. In LQC, the future singularities in the classical phantom models are avoided, instead, by a recollapsing universe, where a bounce universe in an oscillatory regime occurs when the total energy density reaches a Plankton value [28–34]. In the case of constant EoS of DE [35,36], the bounce appears periodically forever, and the Hubble rate $H$, with its derivative $\dot{H}$, are bounded; in the case of a certain form of scalar field models with a variable EoS of DE [29,30,37], the bounce has an infinite frequency of oscillations at a finite time in future, which leads to a diverge of $\dot{H}$, and then the curvature singularity occurs for $R = 6\dot{H} + 12H^2$. The dynamical properties of the phase space are also studied [35,37,38]. In recent years, [32] investigated the inflation in the LQC-modified scalar-tensor theory and [39,40] focused on the generalised EoS in LQC with interactions between DM and DE.

Originating from k-inflation [41–44], the k-essence model could be viewed as a generalization of the quintessence model, which has a canonical Lagrangian $\mathcal{L} = X - V(\phi)$. The k-essence model provides a variety of non-canonical Lagrangian terms, i.e., $\mathcal{L} = F(X, \phi)$ [45–47], such as $\mathcal{L} = F(X)V(\phi)$, $\mathcal{L} = F(X) - V(\phi)$, $\mathcal{L} = V(\phi)f(\frac{F(X)}{V(\phi)})$ and so on, which are used as the pressure of the dark energy [48–52]. The action is described by a single scalar field $\phi$, and a canonical kinetic energy $X \equiv -\frac{1}{2}\partial_\mu\phi\partial^\mu\phi$. Some modified kinetic terms have been discussed, such as $F(X) = KX + LX^2 + \dots$ and $F(X) = \frac{1}{2\alpha-1}((AX)^\alpha - 2\alpha\alpha_0\sqrt{AX})$ [45,48,49,53–57]. The potential $V(\phi)$ takes many forms, such as $V(\phi) \propto \phi^n$, $V(\phi) \propto e^{k\phi}$, $V(\phi) \propto sinh^{-1}(k\phi)$ and so on [58].

From the matter clustering properties, dark matter (DM) and dark energy (DE) are not the same substance; however there is research on the interactions between them, even some nonlinear interaction forms [59–67], which could provide a mechanism by which to generate acceleration and alleviate the coincidence problem. A theoretical explanation of the transition of the DE Eos from $w > -1$ to $w < -1$ has also been presented. With the interaction between DM and DE, $\rho_\phi$ and $\rho_m$ do not separately satisfy independent conservation laws. Furthermore, a cosmological evolution system with DE, DM and unparticle with three kinds of interactions are researched in [68].

In this work, since the k-essence DE models are not widely discussed in the frame of LQC, on the basis of our previous work [69], we research the dynamical stabilities of the phase space of two kinds k-essence DE models in the framework of LQC, especially interactions between DM and DE. The first model is of $\mathcal{L}_1 = F(X)V(\phi) - f(\phi)$, where $F(X) = -\sqrt{X} + X$, $V(\phi) \propto 1/\phi^2$, and $f(\phi) = 0$ [67,69], together with a certain kind of interaction $Q$. Another is of $\mathcal{L}_2 = F(X) - V(\phi)$, where $F(X) = X^\eta$, $V(\phi) \propto \phi^n$ [50,51], together with interaction $Q$. We investigate the possible cosmological behavior of these models in Friedmann–Robertson–Walker–Lemaître (FRWL) spacetime by performing a phase-space and stability analysis. The theory is based on [70,71], judging the stability of the critical points by their eigenvalues, whereas, in this model, for the convenience of calculation, the method using the determinant and trace of the Jacobian matrix of the autonomous differential equations is used [72]. Some cosmological quantities will be calculated for each critical point, such as the dark energy density parameter $\Omega_\phi$, the equation of state (EoS) parameter $w_\phi$ of dark energy, and the deceleration parameter $q$, to compare these with the case in Einstein cosmology without a loop quantum effect and the effect of interaction $Q$. The effect of interaction $Q$ will be shown in the dynamical system for each model in LQC. In addition, for each model, a periodical bounce universe will be formed by setting suitable initial values in the dynamical system, in which both the Hubble rate and its derivative $\dot{H}$ are bounded. That is to say, both the Big Rip singularity and the curvature singularity are

avoided. Throughout this paper, we work with a flat, homogeneous, and isotropic FRWL spacetime with a signature $(-, +, +, +)$ and in units $c = 8\pi G = 1$.

This paper is organized as follows: in the next section, we will discuss the phase space analysis of k-essence dark energy models with Lagrangian $\mathcal{L}_1 = F(X)V(\phi)$ in the LQC framework with interaction $Q = \alpha H \rho_m$, where six critical points will be discussed. In the third section, we will choose another Lagrangian $\mathcal{L}_2 = F(X) - V(\phi)$ in the model and proceed with the discussion using another set of phase variables, where ten critical points are discussed. For both models, the critical points in LQC are in accordance with the EC, which could be viewed as a generalization of EC [51,69]. However, compared with EC, in LQC the evolutions of the cosmological quantities are quite different in the early time universe; additionally, the evolution of the Hubble rate is bounded and oscillates forever. Finally, we close with a few concluding remarks in the fifth section.

## 2. Model I: $\mathcal{L}_1 = F(X)V(\phi)$ in Loop Quantum Cosmology

In a flat universe, the effectively modified Friedmann equation in the framework of LQC is given by [29,30]

$$3H^2 = \rho(1 - \frac{\rho}{\rho_{lc}}),\tag{1}$$

where $H$ is the Hubble parameter, $\rho = \rho_m + \rho_\phi$ is the total energy density and $\rho_\phi$, $\rho_m$ are the energy densities of dark energy and dark matter, respectively. The constant

$$\rho_{lc} = \frac{\sqrt{3}}{16\pi^2\gamma^3 G^2 \hbar} = \frac{4\sqrt{3}}{\gamma^3 \hbar},\tag{2}$$

is the critical loop quantum density, where $\gamma$ is the dimensionless Barbero–Immirzi parameter. Usually, $\rho_{lc} = 1.5$ [29,35], and the LQC goes back to EC when $\rho \ll \rho_{lc}$ or $\rho_{lc} \to \infty$. Considering the interaction between DM and DE, the conservative equation of the total energy density $\dot{\rho} + 3H(p + \rho) = 0$ is represented as

$$\dot{\rho}_\phi + 3H(\rho_\phi + p_\phi) = -Q,\tag{3}$$
$$\dot{\rho}_m + 3H(\rho_m + p_m) = Q.\tag{4}$$

From (1), one can obtain the modified Raychaudhuri equation as

$$\dot{H} = -\frac{1}{2}(\rho_m + \rho_\phi + p_\phi)(1 - 2\frac{\rho}{\rho_{lc}}).\tag{5}$$

In this section, against the background of LQC, we consider a k-essence dark energy model with Lagrangian

$$\mathcal{L}_1 = p_\phi = F(X)V(\phi),\tag{6}$$

as the pressure of the scalar field, while the energy density $\rho_\phi = V(\phi)[2XF'_X - F]$. Then, the density parameter $\Omega_\phi$, the EoS parameters $w_\phi$, $w_{tot}$ and the deceleration parameter $q$ are, respectively, given by

$$\Omega_\phi = \frac{V}{3H^2}[2XF'_X - F],\tag{7}$$
$$w_\phi = \frac{F}{2XF'_X - F},\tag{8}$$
$$w_{tot} = \frac{FV}{3H^2(1-z)^{-1}},\tag{9}$$
$$q = -1 - \frac{\dot{H}}{H^2},\tag{10}$$

where $F_X' \equiv dF/dX$ and $F_{XX}'' \equiv d^2F/dX^2$. And according to $F(X) = -\sqrt{X} + X$, it has $2XF_X' - F = X$.

The equation of motion for the k-essence field is given by

$$\ddot{\phi}\dot{\phi}[F_X' + 2XF_{XX}'']V + \dot{\phi}[2XF_X' - F]V_\phi' + 6XF_X'VH = -Q, \tag{11}$$

where $V_\phi' \equiv dV/d\phi$. Equations (3) and (5) are usually transformed into an autonomous dynamical system when performing the phase-space and stability analysis. By setting the phase variables:

$$x = \dot{\phi}, y = \frac{\sqrt{V}}{\sqrt{3}H}, z = \frac{\rho}{\rho_{lc}}, s = \frac{V_\phi'}{V^{3/2}}, \tag{12}$$

where $0 \leq z \leq 1$ by Equation (1), and using Equation (12), the modified Friedmann Equation (1) and Raychaudhuri Equation (5) become

$$1 = (\Omega_m + y^2[2XF_X' - F])(1 - z), \tag{13}$$

$$\dot{H} = -\frac{1}{2}(\rho_m + V[2XF_X' - F])(1 - 2z). \tag{14}$$

Then, the autonomous dynamical system of the phase variables is:

$$x' = \frac{-1}{F_X' + 2XF_{XX}''}(3xF_X' + \sqrt{3}ys[2XF_X' - F] + \frac{Q}{xVH}), \tag{15}$$

$$y' = \frac{\sqrt{3}}{2}xy^2s + (\frac{3}{2}\frac{y}{1-z} + \frac{3}{2}y^3F)(1 - 2z), \tag{16}$$

$$z' = -3z - 3z(1-z)Fy^2, \tag{17}$$

$$s' = \sqrt{3}\Gamma yx - \frac{3\sqrt{3}}{2}xys^2, \tag{18}$$

where the prime denotes for the derivative to $N = lna$ and the parameter $\Gamma = \frac{V_{\phi\phi}''}{V^2}$. In this work, we chose $V = k\phi^{-2}$ ($k$ is a constant), then $\Gamma = 6k^{-1}$ and $s = -2k^{-\frac{1}{2}}$ simultaneously became constants. As a result, the 4-dim system became a 3-dim system made by $\{x, y, z\}$.

By choosing $F = -\sqrt{X} + X$ and $Q = \alpha H\rho_m$, the autonomous system was:

$$x' = \frac{3\sqrt{2}}{2} - 3x - \frac{\sqrt{3}}{2}x^2ys - \frac{\alpha}{xy(1-z)} + \frac{\alpha}{2}x, \tag{19}$$

$$y' = \frac{\sqrt{3}}{2}xy^2s + (\frac{3}{2}\frac{y}{1-z} + \frac{3}{2}y^3[-\frac{\sqrt{2}}{2}x + \frac{1}{2}x^2])(1 - 2z), \tag{20}$$

$$z' = -3z - 3z(1-z)y^2[-\frac{\sqrt{2}}{2}x + \frac{1}{2}x^2], \tag{21}$$

for $x > 0$; while, for $x < 0$, the system was

$$x' = -\frac{3\sqrt{2}}{2} - 3x - \frac{\sqrt{3}}{2}x^2ys - \frac{\alpha}{xy(1-z)} + \frac{\alpha}{2}x, \tag{22}$$

$$y' = \frac{\sqrt{3}}{2}xy^2s + (\frac{3}{2}\frac{y}{1-z} + \frac{3}{2}y^3[\frac{\sqrt{2}}{2}x + \frac{1}{2}x^2])(1 - 2z), \tag{23}$$

$$z' = -3z - 3z(1-z)y^2[\frac{\sqrt{2}}{2}x + \frac{1}{2}x^2], \tag{24}$$

which has two parameters, i.e., the potential one $s$ and the coupling one $\alpha$. If $z \to 0$, as no loop quantum gravity modification, the dynamical system goes back to the classical

k-essence case, as in [69]. Meanwhile, the cosmological quantities (7)–(10) can be rewritten by the phase variables as:

$$\Omega_\phi = \frac{1}{2}x^2y^2, \tag{25}$$

$$w_\phi = 1 - \sqrt{2}|x|^{-1}, \tag{26}$$

$$w_{tot} = (-\frac{\sqrt{2}}{2}|x| + \frac{1}{2}x^2)y^2(1-z), \tag{27}$$

$$q = \frac{3}{2}[(1-z)^{-1} + y^2(-\frac{\sqrt{2}}{2}|x| + \frac{1}{2}x^2)](1-2z) - 1. \tag{28}$$

The corresponding critical points $\{x_{crit}, y_{crit}, z_{crit}\}$ for $x > 0$ are

$$P_1 = \{\frac{\sqrt{3}}{\sqrt{6}-s}, \frac{\sqrt{6}}{3}(s-\sqrt{6}), 0\}, \tag{29}$$

$$P_2 = \{\frac{\sqrt{3}}{\sqrt{6}+s}, \frac{\sqrt{6}}{3}(s+\sqrt{6}), 0\}, \tag{30}$$

$$P_3 = \{\frac{\sqrt{2}(\alpha-3)^2}{2s^2\alpha+9+\alpha^2-6\alpha}, \frac{\sqrt{6}}{6}\frac{2s^2\alpha+\alpha^2-6\alpha+9}{s(\alpha-3)}, 0\}. \tag{31}$$

While, for $x < 0$, the critical points are:

$$P_4 = \{\frac{-\sqrt{3}}{\sqrt{6}+s}, \frac{-\sqrt{6}}{3}(s+\sqrt{6}), 0\}, \tag{32}$$

$$P_5 = \{\frac{\sqrt{3}}{s-\sqrt{6}}, \frac{\sqrt{6}}{3}(\sqrt{6}-s), 0\}, \tag{33}$$

$$P_6 = \{\frac{-\sqrt{2}(\alpha-3)^2}{2s^2\alpha+9+\alpha^2-6\alpha}, \frac{-\sqrt{6}}{6}\frac{2s^2\alpha+\alpha^2-6\alpha+9}{s(\alpha-3)}, 0\}. \tag{34}$$

The six critical points above are shown in Table 1, in which we also present the necessary conditions for their existences, as well as the corresponding cosmological quantities $\Omega_\phi$, $w_\phi$ and $q$ in form of parameters $s$ and $\alpha$ at each critical point. With these cosmological quantities, we could investigate the final state of the universe and discuss whether an acceleration phase exists. The existences are based on the physical meaning of $x$, $y$ and $z$, and $x > 0$ for $P_1$, $P_2$ and $P_3$; however, $x < 0$ for $P_4$, $P_5$ and $P_6$. They also require $y > 0$ and $z > 0$ according to the definition. Thus, $P_1$ and $P_4$ are excluded by the existence requirement. Meanwhile, $0 \leq w_\phi \leq 1$, and $w_\phi < -1/3$ are needed to accelerate expansion. From the viewpoint of LQC, all critical points have $z_{crit} = 0$, while $x_{crit}$ and $y_{crit}$ are as same as in EC [67,69], which implies that the loop quantum gravity effect vanishes and the EC dominates the whole universe in the late-time universe.

The stability of each point, except $P_1$ and $P_4$, is analysed by the three eigenvalues $\lambda_1$, $\lambda_2$, $\lambda_3$ of the $3 \times 3$ Jacobian matrix as in [29,37,38] for the 3-dim system:

Point $P_2$: $\sqrt{6}s$, $-3 - \frac{\sqrt{6}}{2}s$, $-\sqrt{6}s - 3 + \alpha$,

Point $P_3$: $\alpha - 3$, $\frac{1}{4s(\alpha-3)^2}(-2\alpha^3s + 6\alpha s^3 + 9\alpha^2s - 27s + \sqrt{\Delta})$, $\frac{1}{4s(\alpha-3)^2}(-2\alpha^3s + 6\alpha s^3 + 9\alpha^2s - 27s - \sqrt{\Delta})$,

Point $P_5$: $-\sqrt{6}s$, $-3 + \frac{\sqrt{6}}{2}s$, $+\sqrt{6}s - 3 + \alpha$,

Point $P_6$: $\alpha - 3$, $\frac{1}{4s(\alpha-3)^2}(-2\alpha^3s + 6\alpha s^3 + 9\alpha^2s - 27s + \sqrt{\Delta})$, $\frac{1}{4s(\alpha-3)^2}(-2\alpha^3s + 6\alpha s^3 + 9\alpha^2s - 27s - \sqrt{\Delta})$,

where $\Delta = 36\alpha^2s^6 - 2\alpha^7 + 36\alpha^5s^2 - 108\alpha^3s^4 + 42\alpha^6 - 459\alpha^4s^2 + 648\alpha^2s^4 - 378\alpha^5 + 2268\alpha^3s^2 - 972\alpha s^4 + 1890\alpha^4 - 5346\alpha^2s^2 - 5670\alpha^3 + 5832\alpha s^2 + 10{,}206\alpha^2 - 2187s^2 - 10{,}206\alpha + 4374$.

**Table 1.** The existence and stability conditions for six critical points; the cosmological quantities in the form of parameters $s$ and $\alpha$ at each critical point.

| Name | Existence | Stability | $0 \le \Omega_\phi \le 1$ | $w_\phi < -1/3$ | $q < 0$ |
|---|---|---|---|---|---|
| $P_1$ | none | $0 < s < \sqrt{6}$, $\sqrt{6}s + \alpha - 3 < 0$ | $1$ | $\frac{\sqrt{6}s-3}{3}$ | $\frac{\sqrt{6}s}{2} - 1$ |
| $P_2$ | $\sqrt{6} - s > 0$ | $-\sqrt{6} < s < 0$ $-\sqrt{6}s + \alpha - 3 < 0$ | $1$ | $\frac{\sqrt{6}s+3}{-3}$ | $-\frac{\sqrt{6}s}{2} - 1$ |
| $P_3$ | $(\alpha - 3)^2 > -2s^2\alpha$ $s(\alpha - 3) > 0$ | $\lambda_1 = \alpha - 3 < 0$ $\lambda_2\lambda_3 > 0, \lambda_2 + \lambda_3 < 0$ | $\frac{(\alpha-3)^2}{6s^2}$ | $\frac{-2\alpha s^2}{(\alpha-3)^2}$ | $\frac{1-\alpha}{2}$ |
| $P_4$ | none | $0 < s < \sqrt{6}$, $\sqrt{6}s + \alpha - 3 < 0$ | $1$ | $\frac{\sqrt{6}s-3}{3}$ | $-\frac{\sqrt{6}s}{2} - 1$ |
| $P_5$ | $\sqrt{6} - s > 0$ | $-\sqrt{6} < s < 0$ $-\sqrt{6}s + \alpha - 3 < 0$ | $1$ | $\frac{\sqrt{6}s+3}{-3}$ | $\frac{\sqrt{6}s}{2} - 1$ |
| $P_6$ | $(\alpha - 3)^2 > -2s^2\alpha$ $s(\alpha - 3) > 0$ | $\lambda_1 = \alpha - 3 < 0$ $\lambda_2\lambda_3 > 0, \lambda_2 + \lambda_3 < 0$ | $\frac{(\alpha-3)^2}{6s^2}$ | $\frac{-2\alpha s^2}{(\alpha-3)^2}$ | $\frac{1-\alpha}{2}$ |

Points $P_2$ and $P_5$ each have three real eigenvalues, which implies $\lambda_i < 0$ for stability. However, points $P_3$ and $P_6$, have a real eigenvalue $\lambda_1$ and a pair of quadratic eigenvalues $\lambda_2$ and $\lambda_3$, which indicates the decomposition of an 1-dim subspace and a 2-dim subspace. The stability requires that the single real eigenvalue $\lambda_1 < 0$; meanwhile, the 2-dim subspace requires a determinant larger than zero and trace of less than zero, i.e., $det = \lambda_2 \cdot \lambda_3 > 0$ and $tr = \lambda_2 + \lambda_3 < 0$, without determining whether $\lambda_2$ and $\lambda_3$ are real numbers or complex ones. Additionally, constrained by the existence, stability, $\Omega_\phi$ and the accelerating expansion requirement $w_\phi < -1/3$, the corresponding value ranges for parameters $\alpha$ and $s$ are depicted in Figure 1a for $P_2$ and Figure 2a for $P_3$. Meanwhile, with the observational constrains of $\alpha = 0.009^{+0.013}_{-0.012}$ in [59], $|\alpha| < 0.01$ in [60], and $-1.06 < w_\phi < -0.996$ in [5], together with the expressions of $w_\phi$ for each stable point in Table 1, parameters $\alpha$ and $s$ are constrained in certain values, i.e., $-0.005 < s < 0.073$ for $P_2$ and $0.498(\alpha - 3)^2/\alpha < s^2 < 0.53(\alpha - 3)^2/\alpha$ for $P_3$. To make comparisons with different coupling parameter values $\alpha$, we set $|\alpha| \le 0.02$ in Model I. Since the properties for $x > 0$ are symmetrical to $x < 0$, we only show the evolutional trajectories in the 3-dim phase space around $P_2$ ($s = -0.005$, $\alpha = 0.01$) in Figure 1b and $P_3$ ($s = -16.5$, $\alpha = 0.015$) in Figure 2b. With suitable parameter values, all trajectories around $P_2$ and $P_3$ drop down from $z = 0.1$ to $z_{crit} = 0$, and converge to the attractors $P_2 = (0.709, 1.996, 0)$ and $P_3 = (0.738, 0.142, 0)$ on $x$-$y$ plane, respectively.

Then, we will make two kinds of comparisons: the first one is between LQC and EC from the perspective of trajectories in phase space, and the second one is the effects on the evolution of cosmological quantities for different values of $s$ and $\alpha$. First, compared with the former work in EC [69], there is no difference in the form of $x_c$ and $y_c$ for each critical point in LQC, and $z_{crit} = 0$ for all critical points, which indicates that the final states of the universe are scarcely effected in the LQC. However, the early universe is effected in LQC model, which is shown by the projection of evolutional trajectories from the 3-dim phase space onto $x$-$y$ phase plane for the past, i.e., $lna < 0$, $a < 1$, as in Figure 3, where the trajectories of LQC further separate from the EC ones due to the effect of the LQC phase variable $z$ as $lna \to -\infty$. As a result, the evolution of each cosmological quantity of LQC varies from that of EC in the early universe, but reaches an identical and stable value in the future, as shown in Figures 4 and 5. (It is interesting that the oscillation of $w_\phi$ in Figure 5 is due to the long period spiral around $P_3$, as in Figure 2b; however, $w_\phi$ finally reach an identical and stable value in both LQC and EC for a longer period.) Second, Figure 4 shows that the coupling parameter $\alpha$ and the potential parameter $s$ have little effect on the evolution of the cosmological quantities. From Figure 5, the effects of $\alpha$ and $s$ on the cosmological quantities $\Omega_\phi$ and $q$ are not obvious, but the EoS of DE $w_\phi$ is sensitive to both the coupling parameter $\alpha$ and the potential parameter $s$. It is worth noting that, without the observational constrains on $\alpha$ and $s$, the evolution curves of each cosmological quantity obviously vary with larger values of $\alpha$ and $s$.

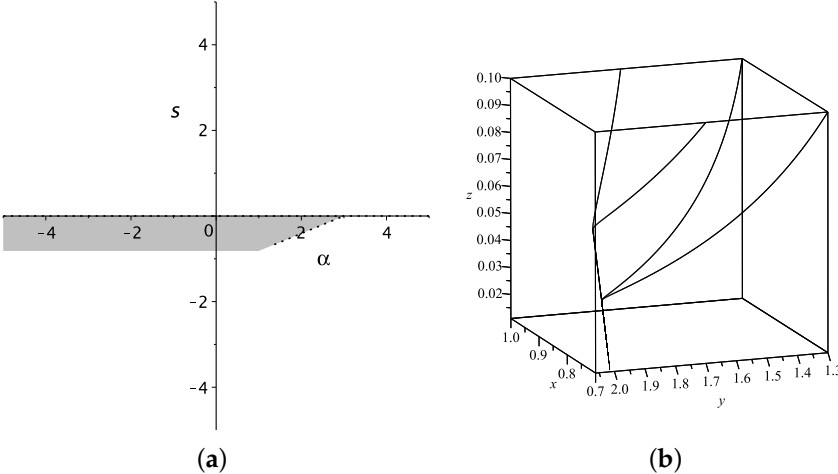

**Figure 1.** (**a**) The value ranges for parameters $s$ and $\alpha$ to create and stablize the critical point $P_2$ when $x > 0$; (**b**) The 3-dim phase space for $s = -0.005$, and $\alpha = 0.01$ around the attractor $P_2 = (0.709, 1.996, 0)$ when $x > 0$.

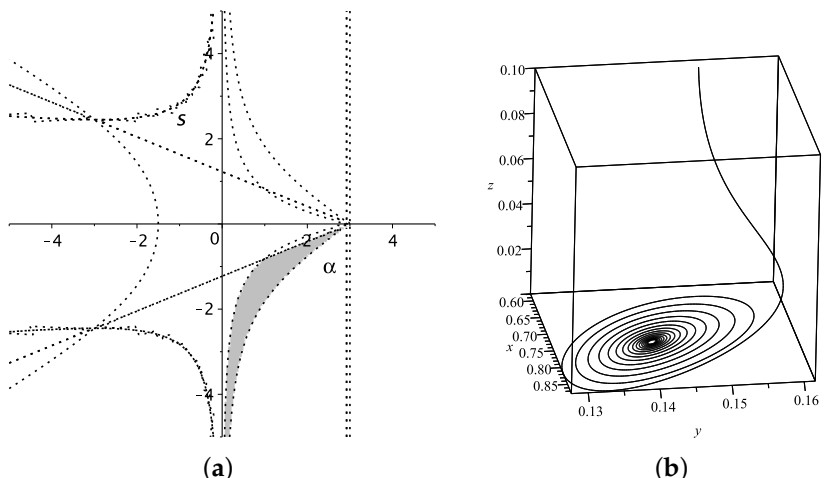

**Figure 2.** (**a**) The value ranges for parameters $s$ and $\alpha$ to create and stabilize the critical point $P_3$ when $x > 0$; (**b**) The 3-dim phase space for $s = -16.5$, and $\alpha = 0.015$ around the attractor $P_3 = (0.738, 0.142, 0)$ when $x > 0$.

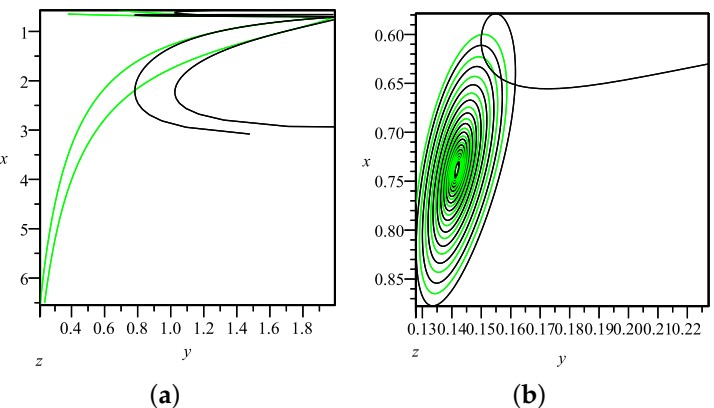

**Figure 3.** The projection of trajectories from 3-dim phase space onto $x$-$y$ plane, and $z$-axis is vertical to the $x$-$y$ plane. (**a**) The phase plane around $P_2 = (0.709, 1.996, 0)$. The green curves denote EC and the black curves denote LQC; (**b**) The phase plane around $P_3 = (0.738, 0.142, 0)$. The green curve denotes EC and the black curve denotes LQC.

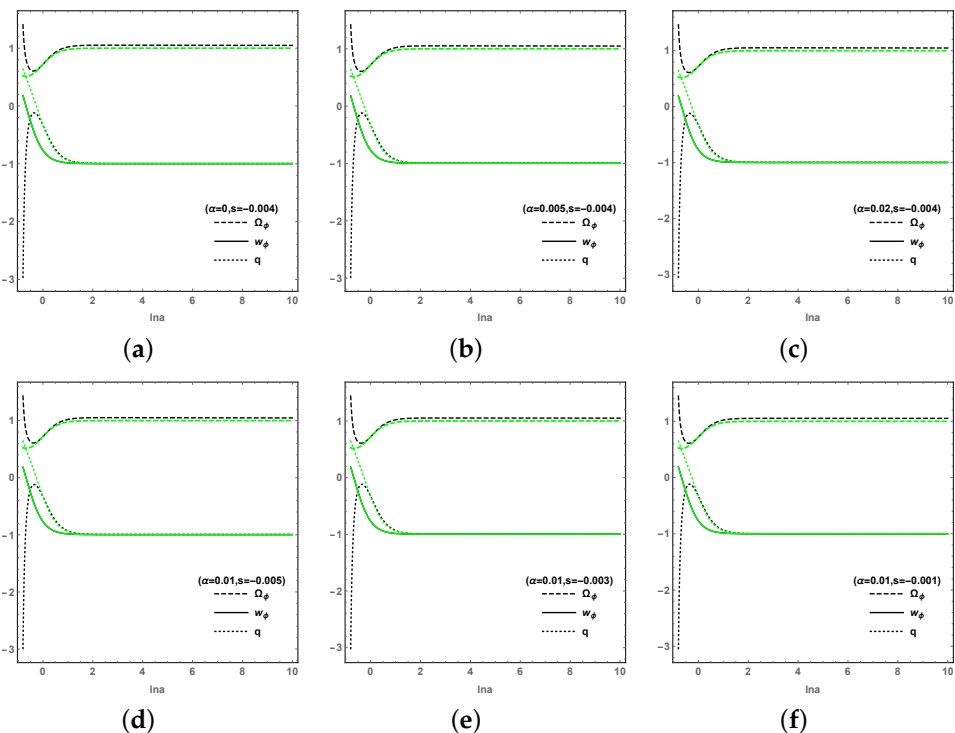

**Figure 4.** The comparison of the evolutions of $\Omega_\phi$, $w_\phi$ and $q$ in EC (green) with the ones in LQC (black) around $P_2$ by the initial condition $x_0 = 0.8, y_0 = 1.5, z_0 = 0.1$. (**a**–**c**) correspond to different coupling parameters $\alpha = 0, 0.005, 0.02$ and fixed potential parameter $s = -0.004$; alternately, (**d**–**f**) correspond to different potential parameters $s = -0.005, -0.003, -0.001$ and fixed coupling parameter $\alpha = 0.01$.

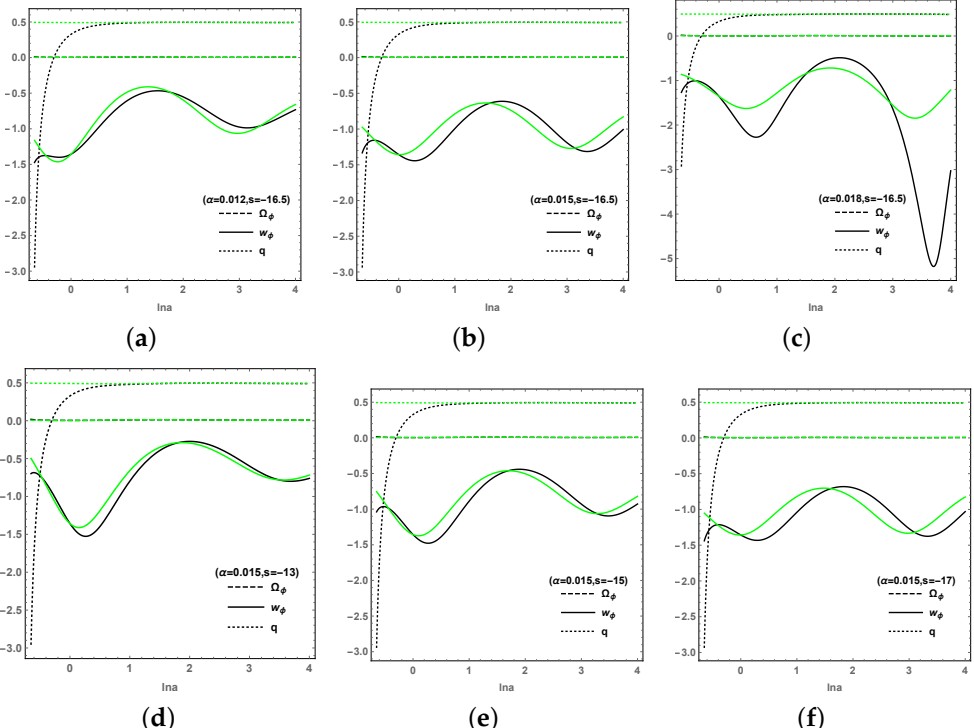

**Figure 5.** The comparison of the evolutions of $\Omega_\phi$, $w_\phi$ and $q$ in EC (green) with those in LQC (black) around $P_2$ by the initial condition $x_0 = 0.6, y_0 = 0.15, z_0 = 0.1$. (**a**–**c**) correspond to different coupling parameters $\alpha = 0.012, 0.015, 0.018$ and fixed potential parameter $s = -16.5$; alternately, (**d**–**f**) correspond to different potential parameters $s = -13, -15, -17$ and fixed coupling parameter $\alpha = 0.015$.

It is well known that the LQC could lead to a bouncing solution [35,36], which helps to avoid singularities. We drew the evolutions of $H$, $\rho_{tot}$ and $\dot{H}$ in Figures 6 and 7 for $P_2$ and $P_3$, respectively, which are based on another autonomous dynamical system composed of the derivative of $\phi$, $\dot{\phi}$, $H$ and $\rho$ with respect to $t$, as in [29,30]:

$$\dot{\phi} = \dot{\phi}, \tag{35}$$

$$\ddot{\phi} = \dot{\phi}^2\phi^{-1} + 3\sqrt{2}H - 6H\dot{\phi} - \frac{\alpha H\rho\phi^2}{k\dot{\phi}} + \frac{1}{2}\alpha H\dot{\phi}, \tag{36}$$

$$\dot{H} = -\frac{1}{2}[\rho + (-\frac{\sqrt{2}}{2}\dot{\phi} + \frac{\dot{\phi}^2}{2})k\phi^{-2}](1 - 2\frac{\rho}{\rho_{lc}}), \tag{37}$$

$$\dot{\rho} = -3H[\rho + (-\frac{\sqrt{2}}{2}\dot{\phi} + \frac{\dot{\phi}^2}{2})k\phi^{-2}]. \tag{38}$$

We set the initial value of $\phi_0$, $\dot{\phi}_0$, $H_0$ and $\rho_0$ under the conditions of the initial loop quantum gravity effect, as $z_0 = \frac{\rho_0}{\rho_{lc}} = 0.01$, i.e., $\rho_0 = 0.015$. Then, $H_0 = 0.07$ from Equation (1). For $P_2$, $k = 16$ as $s = -0.5$ were obtained by the definition; while, $\phi_0 = 20$ and $\dot{\phi}_0 = 0.75$ were obtained for $(x_0, y_0) = (0.75, 1.6)$ around the stable point $P_2$. For another stable point $P_3$, $k = 1/30$ as $s = -11$ by the definition, while $\phi_0 = 9$ and $\dot{\phi}_0 = 0.9$ for $(x_0, y_0) = (0.9, 0.17)$ around the stable point $P_3$. In both cases, the bouncing is terminal, which indicates that the LQC effect acts not only in the early universe but also in future; then, the Big Rip singularity will not appear in future. In addition, the bounce appears periodically forever and the Hubble rate $H$ with its derivative $\dot{H}$ are bounded; then, the curvature singularity is avoided for $R = 6\dot{H} + 12H^2$. From Figures 6b and 7b, when $\rho$ reaches the maximum $\rho_{lc} = 1.5$, the Hubble rate becomes zero for Equation (1), and $\dot{H}$ also reaches its maximum. On the left-hand side of the bouncing, $H < 0$ is the universe expanding backwards, while on the right-hand side, the universe keeps expanding for $H > 0$. It can be concluded that, in future, although each of the cosmological quantities reach a stable value at the stable point, the total energy density $\rho$ oscillates, which leads to the bounce in $H$ and $\dot{H}$ as a recollapse universe.

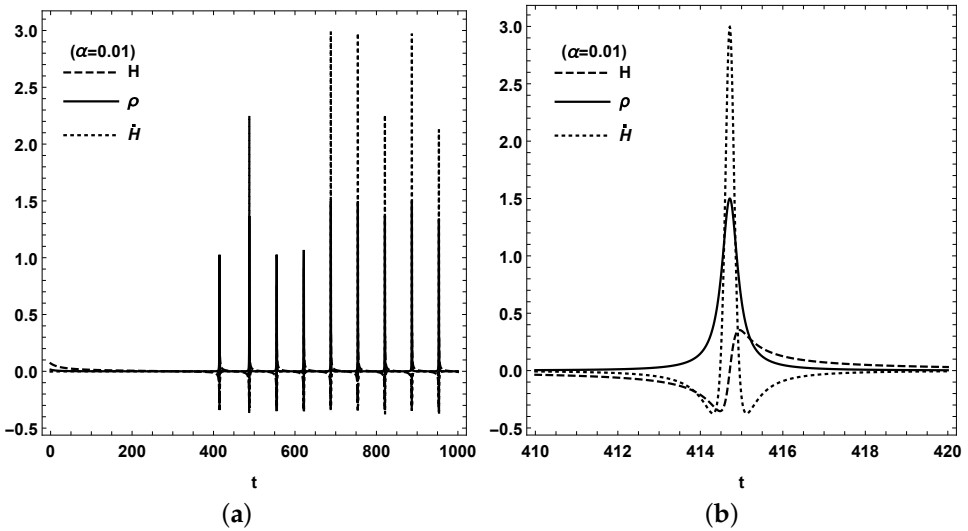

**Figure 6.** (a) The evolution of $H$, $\dot{H}$ and $\rho$ in LQC around $P_2$, by the initial condition of $\phi_0 = 20$, $\dot{\phi}_0 = 0.75$, $H_0 = 0.07$ and $\rho_0 = 0.015$; (b) The local figure in (a) of $t \in [410, 420]$ where bouncing occurs.

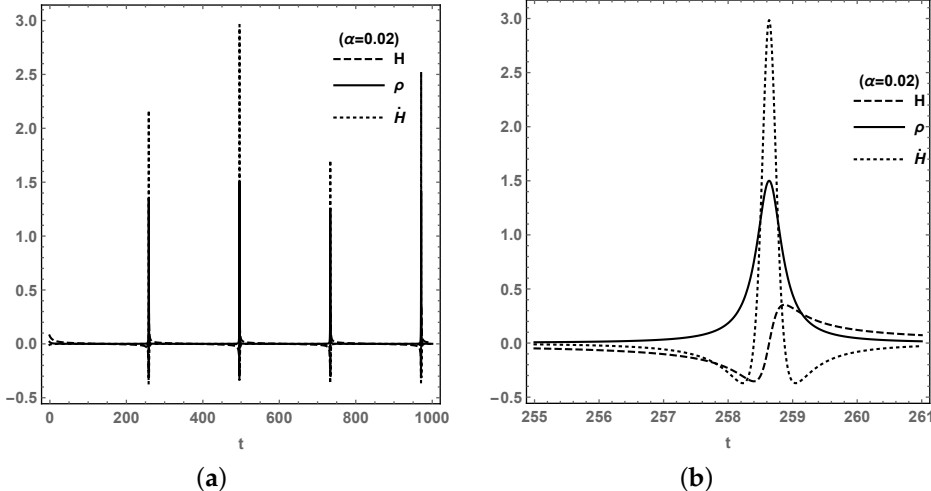

**Figure 7.** (**a**) The evolutions of $H$, $\dot{H}$ and $\rho$ in LQC around $P_3$, by the initial condition of $\phi_0 = 9$, $\dot{\phi}_0 = 0.9$, $H_0 = 0.07$ and $\rho_0 = 0.015$; (**b**) The local figure in (**a**) of $t \in [255, 260]$ where bouncing occurs.

### 3. Model II: $\mathcal{L}_2 = F(X) - V(\phi)$ in Loop Quantum Cosmology

In this section, we consider another kind of k-essence dark energy model with Lagrangian

$$\mathcal{L}_2 = p_\phi = F(X) - V(\phi), \tag{39}$$

then the energy density and the corresponding cosmological quantities, such as the DE density parameter $\Omega_\phi$, the decelerating parameter $q$, and the EoS parameters, are given by

$$\rho_\phi = V(\phi) + [2XF'_X - F], \tag{40}$$

$$\Omega_\phi = (2XF'_X - F + V)/(3H^2), \tag{41}$$

$$q = -1 + \frac{3}{2}(\frac{2XF'_X}{3H^2} + \Omega_m)(1 - 2\frac{\rho}{\rho_{lc}}), \tag{42}$$

$$w_{tot} = \frac{F - V}{3H^2}(1 - \frac{\rho}{\rho_{lc}}), \tag{43}$$

$$w_\phi = \frac{F - V}{2XF'_X - F + V}. \tag{44}$$

According to the method in [50,51], in (40) we let $\rho_k = 2XF'_X - F$, by setting the phase variables as

$$x = \sqrt{\frac{\rho_k}{3H^2}}, \ y = \sqrt{\frac{V}{3H^2}}, \ z = \frac{\rho}{\rho_{lc}}, \ \sigma = \frac{-1}{\sqrt{3\rho_k}}\frac{V'_\phi\dot{\phi}}{V}. \tag{45}$$

Then, the general autonomous dynamical system of the phase variables with three parameters $\gamma_k$, $\Xi$, $\Gamma$ and interaction $Q$ is as follows,

$$x' = \frac{3x}{2}(\frac{1}{1-z} - x^2 - y^2 + \gamma_k x^2)(1 - 2z) - \frac{3}{2}(\gamma_k x - \sigma y^2) - \frac{Q}{6H^3}, \tag{46}$$

$$y' = \frac{3}{2}y[(\frac{1}{1-z} - x^2 - y^2) + \gamma_k x^2](1 - 2z) - \frac{3}{2}xy\sigma, \tag{47}$$

$$z' = -3z + 3z(1 - z)(x^2 - \gamma_k x^2 + y^2), \tag{48}$$

$$\sigma' = 3\sigma^2 x(1 - \Gamma) + \frac{3}{2}\sigma\gamma_k - \frac{3}{2}\sigma^2 x^{-1}y^2 - \frac{Q}{6H^3} - \frac{3\sigma}{1 + 2\Xi}(1 - \sigma x^{-1}y^2\gamma_k^{-1}), \tag{49}$$

where $\gamma_k = \frac{2XF'_X}{2XF'_X - F}$, $\Xi = \frac{XF''_{XX}}{F'_X}$, $\Gamma = \frac{V''_{\phi\phi}V}{V^2_\phi}$. In [73], this provides a novel setting of phase variables based on a special $F(X)$, as in [57]. Whereas, in this work, by setting $F(X) = AX^\eta$ and $V = B\phi^n$, the three variables $\gamma_k = \frac{2\eta}{2\eta-1}$, $\Xi = \eta - 1$ and $\Gamma = \frac{n-1}{n}$ become constants. The interaction is $Q = \alpha H\rho_m$, as in model I, then $\frac{Q}{6H^3} = \frac{\alpha}{2}(\frac{1}{1-z} - x^2 - y^2)$ in (46). For $z \to 0$, the above equations go back to the case in Einstein cosmology, as in [50,51]. By solving the equations above, we obtain eight critical points. For existence, $x > 0$, $y > 0$ and $0 < z < 1$ are required by definition. It is also constrained by $0 \leq \Omega_\phi \leq 1$ for physical meaning in the late-time universe, and $w_\phi < -1/3$ for accelerating expansion. The details are listed in Table 2. In this case, the cosmological quantities expressed by the phase variables are as follows :

$$\Omega_\phi = x^2 + y^2, \tag{50}$$

$$q = -1 + \frac{3}{2}\left(\frac{1}{1-z} - x^2 - y^2 + \gamma_k x^2\right)(1 - 2z), \tag{51}$$

$$w_{tot} = (\gamma_k x^2 - x^2 - y^2)(1 - z), \tag{52}$$

$$w_\phi = \frac{-x^2 - y^2 + \gamma_k x^2}{x^2 + y^2}. \tag{53}$$

The eight critical points for $\mathcal{L}_2 = AX^\eta - B\phi^n$ are:

$$P_a = \{0, 0, 0, 0\}, \; for \; \alpha = 0, \tag{54}$$

$$P_b = \{1, 0, 0, 0\}, \tag{55}$$

$$P_c = \{0, 1, 0, 0\}, \tag{56}$$

$$P_d = \{1, 0, 0, \frac{2 - \gamma_k}{2\Gamma - 2}\}, \tag{57}$$

$$P_{e1} = \{\frac{\alpha}{3(1 - \gamma_k)}, 0, 0, \frac{9\gamma_k^3 - 36\gamma_k^2 + 45\gamma_k - 18 + \sqrt{\Delta_e}}{12\alpha(\Gamma\gamma_k - \Gamma - \gamma_k + 1)}\}, \tag{58}$$

$$P_{e2} = \{\frac{\alpha}{3(1 - \gamma_k)}, 0, 0, \frac{9\gamma_k^3 - 36\gamma_k^2 + 45\gamma_k - 18 - \sqrt{\Delta_e}}{12\alpha(\Gamma\gamma_k - \Gamma - \gamma_k + 1)}\}, \tag{59}$$

$$P_{f1} = \{\frac{3}{\alpha}, 0, \frac{\alpha^2 + 9\gamma_k - 9}{9(\gamma_k - 1)}, \frac{-\alpha\gamma_k + 2\alpha + \sqrt{\Delta_f}}{12(\Gamma - 1)}\}, \tag{60}$$

$$P_{f2} = \{\frac{3}{\alpha}, 0, \frac{\alpha^2 + 9\gamma_k - 9}{9(\gamma_k - 1)}, \frac{-\alpha\gamma_k + 2\alpha - \sqrt{\Delta_f}}{12(\Gamma - 1)}\}, \tag{61}$$

where $\Delta_e = -8\alpha^2(\gamma_k - 1)(\Gamma - 1)(\alpha + 3\gamma_k - 3)(\alpha + 3 - 3\gamma_k) + 81(\gamma_k - 2)^2(\gamma_k - 1)^4$ and $\Delta_f = 72\gamma_k(\Gamma - 1) + \alpha^2(\gamma_k - 2)^2$. $P_a$ only exists for the coupling parameter $\alpha = 0$. It is very interesting that, under the action of the interaction $Q$, the system has four extra critical points $P_{e1}$, $P_{e2}$, $P_{f1}$ and $P_{f2}$, and they are only for $\alpha \neq 0$ as non-zero interaction between DE and DM. From the perspective of LQC, unlike other critical points, this has $z_{crit} \neq 0$ in $P_{f1}$ and $P_{f2}$, which indicates that the loop quantum effect still exists in the final states, corresponding to these two points. However, from the requirement $0 < z < 1$, the existences of $P_{f1}$ and $P_{f2}$ contradict the physical meaning. That is to say, $P_{f1}$ and $P_{f2}$ are out of discussion, though $P_{f1}$ is mathematically stable, as discussed later.

The stability of the dynamical system around each critical point is analysed by the eigenvalues $\lambda_1$, $\lambda_2$, $\lambda_3$ and $\lambda_4$ of the $4 \times 4$ Jacobian matrix, as in [38]:

Point $P_a$: $\frac{3}{2}, -3, \frac{3-3\gamma_k}{2}, \frac{6-3\gamma_k}{2}$;

Point $P_b$: $\alpha + 3\gamma_k - 3, 3\gamma_k/2, -3\gamma_k, -3\gamma_k/2 + 3$;

Point $P_c$: $-3, 0, \frac{3}{2}\frac{-\gamma_k^2 + 2\gamma_k - 2 + \sqrt{3\gamma_k^2 - 6\gamma_k + 4}}{2\gamma_k}, \frac{3}{2}\frac{-\gamma_k^2 + 2\gamma_k - 2 - \sqrt{3\gamma_k^2 - 6\gamma_k + 4}}{2\gamma_k}$;

Point $P_d$: $\alpha + 3\gamma_k - 3, 3(2\Gamma\gamma_k - \gamma_k - 2)/(4\Gamma - 4), -3\gamma_k, 3\gamma_k/2 - 3$;

Point $P_{e1}$: $\frac{\alpha^2 - 9\gamma_k^2 + 18\gamma_k - 9}{6(\gamma_k - 1)}, \frac{\alpha^2 + 9\gamma_k - 9}{-3(\gamma_k - 1)}, \frac{\sqrt{\Delta_e}}{6(\gamma_k - 1)^2}, \frac{(\gamma_k - 1)[4\alpha^2(\Gamma - 1) + 9(\gamma_k - 1)(4\Gamma + \gamma_k - 6)] + \sqrt{\Delta_e}}{24(\gamma_k - 1)^2(\Gamma - 1)}$;

Point $P_{e2}$: $\frac{\alpha^2-9\gamma_k^2+18\gamma_k-9}{6(\gamma_k-1)}$, $\frac{\alpha^2+9\gamma_k-9}{-3(\gamma_k-1)}$, $\frac{-\sqrt{\Delta_e}}{6(\gamma_k-1)^2}$, $\frac{(\gamma_k-1)[4\alpha^2(\Gamma-1)+9(\gamma_k-1)(4\Gamma+\gamma_k-6)]-\sqrt{\Delta_e}}{24(\gamma_k-1)^2(\Gamma-1)}$;

Point $P_{f1}$: $\frac{3}{4\alpha}(-\alpha\gamma_k+\sqrt{\Delta_f})$, $\frac{3}{4\alpha}(-\alpha\gamma_k-\sqrt{\Delta_f})$, $-\frac{3}{2\alpha}\sqrt{\Delta_f}$, $\frac{3(\alpha\gamma_k-2\alpha-\sqrt{\Delta_f})}{8\alpha(\Gamma-1)}$;

Point $P_{f2}$: $\frac{3}{4\alpha}(-\alpha\gamma_k+\sqrt{\Delta_f})$, $\frac{3}{4\alpha}(-\alpha\gamma_k-\sqrt{\Delta_f})$, $\frac{3}{2\alpha}\sqrt{\Delta_f}$, $\frac{3(\alpha\gamma_k-2\alpha+\sqrt{\Delta_f})}{8\alpha(\Gamma-1)}$.

Among those critical points, only $P_d$ and $P_{e2}$ satisfy both the existence and stability properties ($x_{crit} \sim y_{crit} \sim z_{crit} \sim \sigma_{crit}$ is supposed to be equivalently infinitesimal in the calculation of the eigenvalues of $P_a$ and $P_c$). We plotted the value region for parameters $\alpha$, $\gamma_k$ and $\Gamma$ in Figures 8a and 9a, together with the dynamical evolution in the 3-dim phase space of $\{x, y, z\}$ with each initial points and parameter values in Figures 8b and 9b. In Model II, the stability condition and physical properties of some parameters move the value range of the parameter $\alpha$ out of the constrain $|\alpha| \leq 0.02$ for some stable points. Since $P_d$ is not effected by the coupling parameter $\alpha$, the trajectories of $\alpha = -1$ and $\alpha = -2$ start from the same initial point and then converge to the same stable point $P_d = (1, 0, 0, \frac{30}{7})$, but the process differs according to the effect of interaction. $\Omega_\phi = 1$ means the universe is dominated by DE, and $(x_d = 1, y_d = 0)$ means the DE is composed by kinetics totally. As $P_{e2}$ is effected by the coupling parameter $\alpha$, the trajectories with $\alpha = -0.01$ and $\alpha = -0.3$ start from the same initial point but converge to different stable points. The DE density parameter $\Omega_\phi = \frac{\alpha^2}{9(\gamma_k-1)^2}$ is affected by the interaction between DE and DM, i.e., the universe is composed of both DE and DM.

**Table 2.** The existence and stability conditions for eight critical points, the cosmological quantities, and the range of the parameters $\gamma_k$, $\Gamma$ and $\alpha$ for acceleration.

| Name | Existence | Stability | $0 \leq \Omega_\phi \leq 1$ | $w_\phi < -1/3$ | $q < 0$ |
|---|---|---|---|---|---|
| $P_a$ | $\alpha = 0$ | unstable | $0$ | none | $\frac{3}{2}$ |
| $P_b$ | always | unstable | $1$ | $\gamma_k - 1$ | $\frac{3\gamma_k}{2} - 1$ |
| $P_c$ | always | unstable | $1$ | $-1$ | $-1$ |
| $P_d$ | always | $0 < \gamma_k < 2, \alpha + 3\gamma_k < 3$ <br> $1 < \Gamma < 1/\gamma_k + 1/2$ | $1$ | $\gamma_k - 1$ | $\frac{3\gamma_k}{2} - 1$ |
| $P_{e1}$ | $\frac{\alpha}{1-\gamma_k} > 0$ | unstable | $\frac{\alpha^2}{9(\gamma_k-1)^2}$ | $\gamma_k - 1$ | $\frac{\alpha^2+3\gamma_k-3}{6(\gamma_k-1)}$ |
| $P_{e2}$ | $\frac{\alpha}{1-\gamma_k} > 0$ | $\lambda_1 < 0, \lambda_2 < 0, \lambda_3 < 0, \lambda_4 < 0$ | $\frac{\alpha^2}{9(\gamma_k-1)^2}$ | $\gamma_k - 1$ | $\frac{\alpha^2+3\gamma_k-3}{6(\gamma_k-1)}$ |
| $P_{f1}$ | none | stable | $\frac{9}{\alpha^2}$ | $\gamma_k - 1$ | $-1$ |
| $P_{f2}$ | none | unstable | $\frac{9}{\alpha^2}$ | $\gamma_k - 1$ | $-1$ |

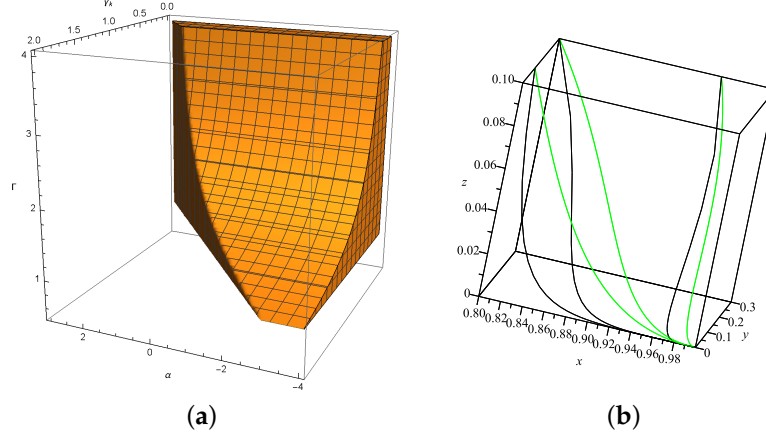

(a)    (b)

**Figure 8.** (**a**) The value ranges for parameters $\alpha$, $\gamma_k$ and $\Gamma$ to create and stabilize the critical point $P_d$; (**b**) The 3-dim phase space for fixed $\gamma_k = \frac{8}{7}$ and $\Gamma = \frac{11}{10}$, corresponding to two different values $\alpha = -1$ (black) and $\alpha = -2$ (green) around $P_d = (1, 0, 0, \frac{30}{7})$.

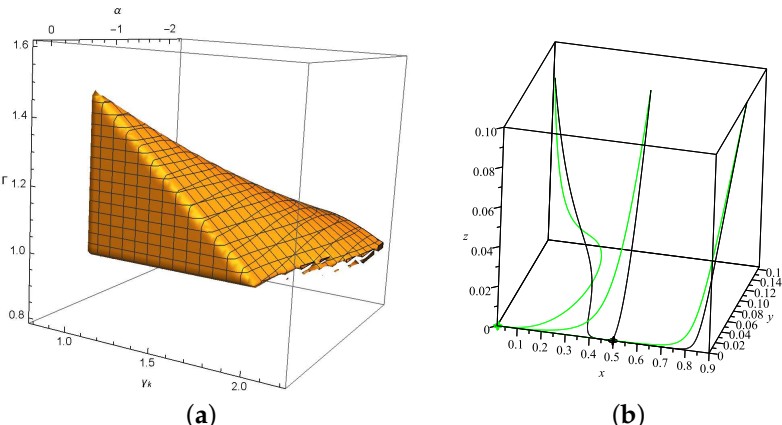

**Figure 9.** (**a**) The value ranges for parameters $\alpha$, $\gamma_k$ and $\Gamma$ to create and stabilize the critical point $P_{e2}$; (**b**) The 3-dim phase space around $P_{e2}$ with different coupling parameter $\alpha$ by fixing $\gamma_k = \frac{6}{5}$ and $\Gamma = \frac{6}{5}$. When $\alpha = -0.3$, the black trajectories converge to $P_{e2} = (\frac{1}{2}, 0, 0, 4)$, while for $\alpha = -0.01$, the green trajectories converge to $P_{e2} = (\frac{1}{60}, 0, 0, 120)$.

By using the symmetry analysis in [50,51] for specific Lagrangian $\mathcal{L} = AX^\eta - B\phi^n$, the 3-dim dynamical system is reduced to a 2-dim one in EC. In this work, the analogous analysis acts in the frame of LQC to reduce the 4-dim system to a 3-dim one. With $dN = dlna$ as an independent variable instesd of $dt$, the equations of $H^2$, $\dot{H}$, $\dot{\rho}_m$ and $\dot{\rho}_\phi$ are rewritten as

$$H^2 = \frac{1}{3}[(2\eta - 1)AX^\eta + B\phi^n + \rho_m](1 - \frac{(2\eta - 1)AX^\eta + B\phi^n + \rho_m}{\rho_{lc}}), \qquad (62)$$

$$H\frac{dH}{dN} = \frac{-1}{2}[2\eta AX^\eta + \rho_m](1 - 2\frac{(2\eta - 1)AX^\eta + B\phi^n + \rho_m}{\rho_{lc}}), \qquad (63)$$

$$\frac{d\rho_m}{dN} = -3\rho_m + \alpha\rho_m, \qquad (64)$$

$$\frac{d}{dN}[(2\eta - 1)AX^\eta + B\phi^n] = -6\eta AX^\eta - \alpha\rho_m. \qquad (65)$$

If we consider $\phi$, $X$, $\rho_m$ and $\rho_{lc}$ as the independent variables ($\rho_\phi$ is composed of $\phi$ and $X$), it has the transformation: $\phi \to \xi^{2\eta}\phi$, $X \to \xi^{2n}X$, $\rho_m \to \xi^{2n\eta}\rho_m$, $\rho_{lc} \to \xi^{2n\eta}\rho_{lc}$. Since $X = \frac{1}{2}(H\frac{d\phi}{dN})^2$, by the transformation above, the Hubble parameter should hold the transformation $H \to \xi^{n-2\eta}H$. However, to leave Equation (62) invariant requires $H \to \xi^{n\eta}H$. The relation of $n$ and $\eta$ can be written as $\eta = \frac{n}{2+n}$, then $\gamma_k = \frac{2\eta}{2\eta - 1} = \frac{2n}{n-2}$ as a result; further, the symmetry allows for a reduction in the dynamical system from four variables into three, as follows:

$$\sigma = -nB^{\frac{1}{n}}\sqrt{\frac{2}{3}}(A(2\eta - 1))^{\frac{-1}{2\eta}}(\frac{x}{y})^{\frac{2}{n}} = s(\frac{x}{y})^{\frac{2}{n}}, \qquad (66)$$

where $s = -nB^{\frac{1}{n}}\sqrt{\frac{2}{3}}(A(2\eta - 1))^{\frac{-1}{2\eta}}$. As a result, the system (46) reduced to the following 3-dim form with three parameters of $\gamma_k$, $s$ and coupling parameter $\alpha$ as follows:

$$x' = \frac{3x}{2}(\frac{1}{1-z} - x^2 - y^2 + \gamma_k x^2)(1 - 2z) - \frac{3}{2}\gamma_k x + \frac{3}{2}sxy(y/x)^{\frac{2}{\gamma_k}} - \frac{\alpha}{2}(\frac{1}{1-z} - x^2 - y^2), \quad (67)$$

$$y' = \frac{3}{2}y[(\frac{1}{1-z} - x^2 - y^2) + \gamma_k x^2](1 - 2z) - \frac{3}{2}sx^2(y/x)^{\frac{2}{\gamma_k}}, \qquad (68)$$

$$z' = -3z + 3z(1 - z)(x^2 - \gamma_k x^2 + y^2). \qquad (69)$$

Then, there are two new critical points $P_g$ and $P_h$:

$$P_g = \{\frac{\sigma}{\gamma_k}, \sqrt{1 - \frac{\sigma^2}{\gamma_k^2}}, 0\}, \tag{70}$$

$$P_h = \{\frac{3}{3\sigma + \alpha}, \frac{\sqrt{\alpha^2 + 3\alpha\sigma + 9\gamma_k - 9}}{3\sigma + \alpha}, 0\} \tag{71}$$

$z_{crit} = 0$ is found in these two critical points, which indicates that the LQC effect disappears in the late-time universe. However, for both of the critical points containing $\sigma$ with $x/y$ inside, the expressions cannot be analytical. As a result, the stability analysis by the Jacobian matrix cannot proceed (the interaction $Q = \alpha H\rho_m$ diminishes the coupling parameter $\alpha$ by substituting $P_g$ into (67)), instead we will take certain parameter values for later calculation. Using the expression (70), $P_g$ is not effected by the coupling parameter $\alpha$, while the interaction effects $P_h$, as in (71). These two critical points are in accordance with [51], in EC for the no interaction case, i.e., $\alpha = 0$. In LQC, by setting $\alpha = 0$, $P_h$ is lucky to have the analytical form $P_{h0}$, which is expressed by parameters $\gamma_k$ and $s$, by substituting $x_h = 1/\sigma$ and $y_h = \sqrt{\gamma_k - 1}/\sigma$ into (66):

$$P_{h0} = \{1/\sigma, \sqrt{\gamma_k - 1}/\sigma, 0\} = \{s^{-1}(\gamma_k - 1)^{\frac{\gamma_k - 2}{2\gamma_k}}, s^{-1}(\gamma_k - 1)^{\frac{\gamma_k - 1}{\gamma_k}}, 0\} \tag{72}$$

With the analytical form of $P_{h0}$, the stability can be discussed using the eigenvalues of the $3 \times 3$ Jacobian matrix of the system, as shown below:

$$P_{h0}: -3, \frac{3}{4}s^{-1}(\gamma_k - 1)^{-2/\gamma_k}(-s(\gamma_k - 1)^{2/\gamma_k} + \sqrt{\Delta_{h0}}), \frac{3}{4}s^{-1}(\gamma_k - 1)^{-2/\gamma_k}(-s(\gamma_k - 1)^{2/\gamma_k} - \sqrt{\Delta_{h0}}),$$

in which $\Delta_{h0} = -(\gamma_k - 1)^{2/\gamma_k}[8\gamma_k s^2(\gamma_k - 1)^{2/\gamma_k} - 9s^2(\gamma_k - 1)^{2/\gamma_k} - 8\gamma_k(\gamma_k - 1)^2]$. We calculated the value ranges of parameters $\gamma_k$ and $\alpha$ under the condition of existence, physical meanings and stabilities for $P_{h0}$, which are listed in Table 3 and depicted in Figure 10a. By setting $s = 3$ and $\gamma_k = \frac{3}{2}$, from Figure 10b, which shows that $P_{h0} = (\frac{\sqrt[6]{2}}{3}, \frac{\sqrt[3]{4}}{6}, 0) \approx (0.374, 0.265, 0)$ has the same stability for both EC and LQC, which indicates that the late-time universe of LQC is in accordance with EC and the loop quantum effects vanish in the late-time universe.

**Table 3.** The existence and stability conditions for three critical points under the symmetric analysis, the cosmological quantities in form of the parameters $\gamma_k$, $\Gamma$ and $\alpha$ .

| Name | Existence | Stability | $\Omega_\phi$ | $w_{tot}$ | $w_\phi$ | $q$ |
|------|-----------|-----------|---------------|-----------|----------|-----|
| $P_g$ | $\sigma/\gamma_k > 0$ | $\gamma_k^2 - \sigma^2 > 0,$ <br> $\alpha\sigma + 3\sigma^2 - 3\gamma_k < 0$ | $1$ | $\frac{\sigma^2 - \gamma_k}{\gamma_k}$ | $\frac{\sigma^2 - \gamma_k}{\gamma_k}$ | $\frac{3\sigma^2 - 2\gamma_k}{2\gamma_k}$ |
| $P_h$ | $\gamma_k > 1, s > 0$ | $\alpha\sigma + 3\sigma^2 - 3\gamma_k > 0,$ <br> $\alpha\gamma_k + 3\gamma_k\sigma - 3\sigma > 0$ | $\frac{\gamma_k}{\sigma^2}$ | $0$ | $0$ | $1/2$ |
| $P_{h0}$ | $\alpha = 0,$ <br> $\gamma_k > 1, s > 0$ | $s^{-2}\gamma_k(\gamma_k - 1)^{\frac{(\gamma_k - 2)}{\gamma_k}} < 1$ | $\frac{\gamma_k}{\sigma^2}$ | $0$ | $0$ | $1/2$ |

After discussing the critical point $P_{h0}$ of the analytical form above, we research the general forms $P_g$ and $P_h$, which are non-analytical, below. Here, we set $\eta = 2$, $n = -4$ and $s = 4$, i.e., $F(X) - V(\phi) = AX^2 - B\phi^{-4}$, as in [51], to compare EC and LQC. In this case, $\gamma_k = \frac{4}{3}$ and $AB = \frac{4M_{pl}^4}{27}$; then, $\alpha$ becomes the only parameter. Numerically, point $P_g = (x_g, y_g, z_g)$ with the eigenvalues $\lambda_i$ behaves as follows:

$$x_g \approx 0.994, \quad y_g \approx 0.109, \quad z_g = 0.$$

$$\lambda_1 \approx -3.95, \quad \lambda_2 \approx 0.95 + 0.99\alpha, \quad \lambda_3 \approx -1.02.$$

Compared with [51] in EC, $x_g$ and $y_g$ are the same as the ones in EC and, in this work, the stability condition requires $\alpha < -0.958$ to make the critical point stable by $\lambda_2 < 0$; while, in [51] $\alpha = 0$ of no interaction means that this point is a saddle point. Figure 11 shows the effect of coupling parameter $\alpha$. Although $\alpha$ has no effects on the location of the critical point $P_g$, it effects the stability, i.e., from Figure 11a–c $P_g$ becomes a stable point from a saddle point by changing the value of the coupling parameter $\alpha$. Conversely, for $P_h$, by setting $\alpha = 0$, this is the stable point $P_{h0} = (\sqrt[4]{3}/4, \sqrt[4]{27}/12, 0)$, in accordance with [51]in EC; then, by changing $\alpha$ from 0 to $-1.6$, it becomes an unstable point. It can be concluded that coupling parameter $\alpha$ effects the stability of the critical points $P_g$ and $P_h$. Further, from Table 2, the critical points $P_g$ and $P_h$ cannot be stable simultaneously under the stability condition.

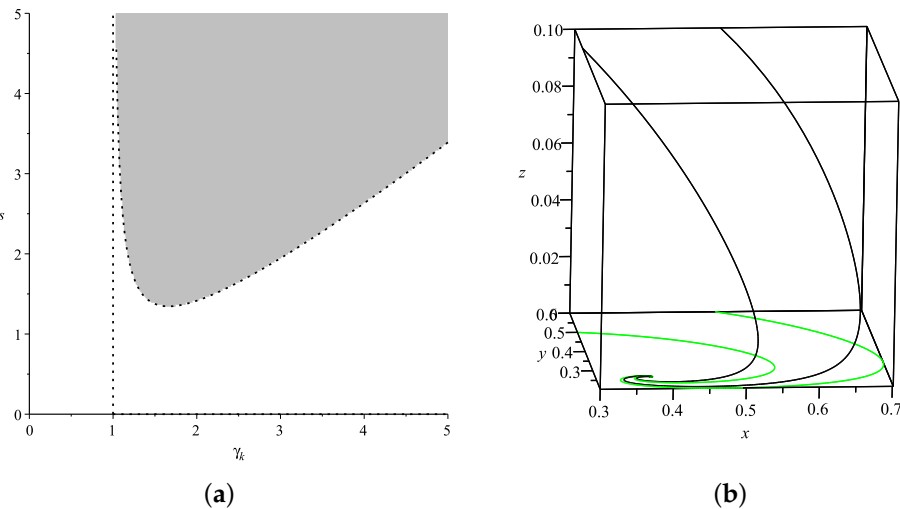

**Figure 10.** (a) Parameter region for $P_{h0}$; (b) The phase space around $P_{h0} = (\frac{\sqrt[6]{2}}{3}, \frac{\sqrt[3]{4}}{6}, 0) \approx (0.374, 0.265, 0)$ with $s = 3$ and $\gamma_k = \frac{3}{2}$ for EC(green) and LQC(black).

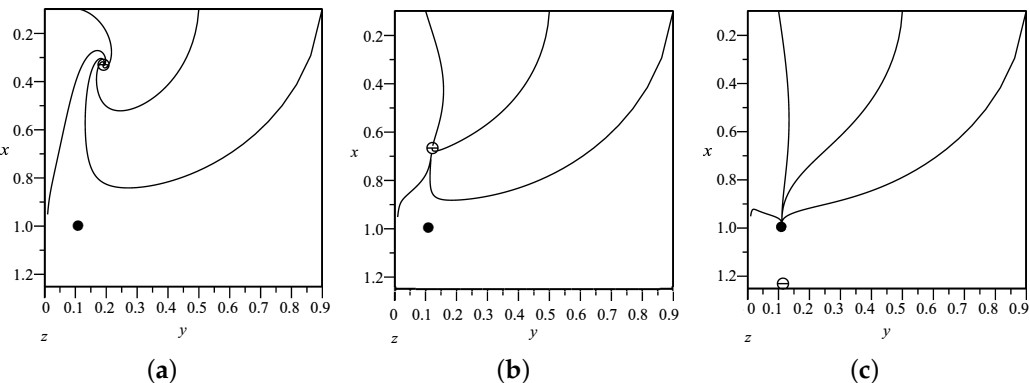

**Figure 11.** The stabilities corresponding to different values of coupling parameter $\alpha$, which are shown by the projection from a 3-dim phase space onto to 2-dim $x$-$y$ plane, and $z$-axis is vertical to the $x$-$y$ plane. $P_g = (0.994, 0.109, 0)$ is displayed by a solid circle. (a) With $\alpha = 0$, trajectories converge to stable point $P_h = (0.329, 0.190, 0)$, just as [51] in EC; (b) With $\alpha = -0.6 > -0.985$, trajectories still converge to stable point $P_h = (0.666, 0.121, 0)$; (c) With $\alpha = -1.2 < -0.985$, all trajectories converge to $P_g$ which is a stable point in this case, but $P_h = (1.231, 0.113, 0)$ becomes unstable.

Based on all four stable points $P_d$, $P_{e2}$, $P_g$ and $P_h$ in model II above, to delineate between the effect of LQC and classical EC, the evolution trajectories for both LQC and EC are projected onto the $x$-$y$ phase plane, as shown in Figures 12a, 13a, 14a and 15a. It can be seen that the dynamical evolutionary trajectories for both LQC and EC, starting from the same initial points around stable points, comply with each other in the late-time

universe. However, the trajectories for the early universe as $lna < 0$ bifurcate from each other. As a result, the values of cosmological quantities in LQC are varied from EC in the early universe, as the loop quantum gravity effect is obvious; however, for the late-time universe with the diminishing of loop quantum effect, the evolutions of the cosmological quantities show no difference from LQC to EC, as depicted in Figures 12b, 13b, 14b and 15b.

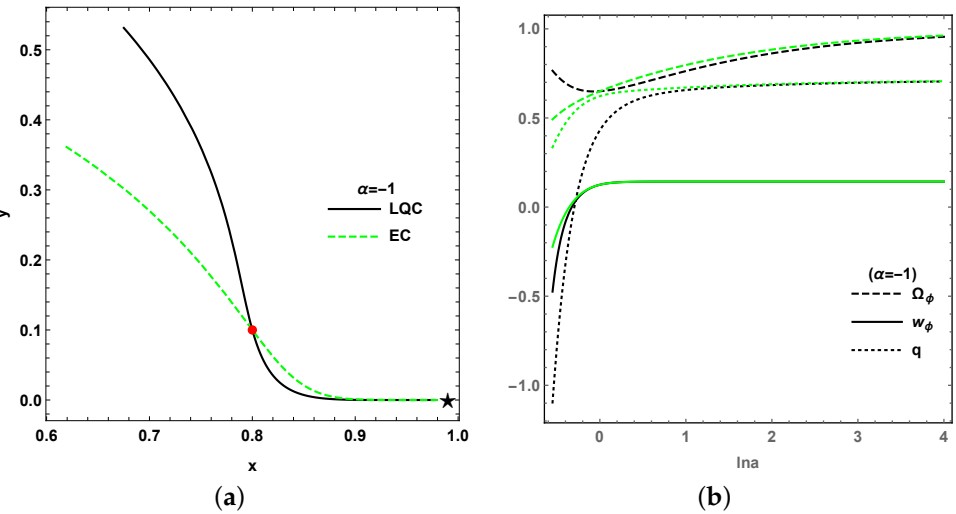

**Figure 12.** (**a**) The $x$-$y$ phase plane projected from 4-dim system of $\{x,y,z,\sigma\}$ for both LQC(black) and EC(green dash) around $P_d$, corresponding to the parameters of $\gamma_k = \frac{8}{7}$, $\Gamma = \frac{11}{10}$ and $\alpha = -1$. The initial points are $(0.8, 0.1, 0.1, 4)$ for LQC, and $(0.8, 0.1, 0, 4)$ for EC. The star stands for the final state of the evolution, while the solid circle stands for the initial point. (**b**) the evolution of the cosmological quantities of $\Omega_\phi$, $w_\phi$ and $q$ for both LQC(black) and EC(green) cases around $P_d$.

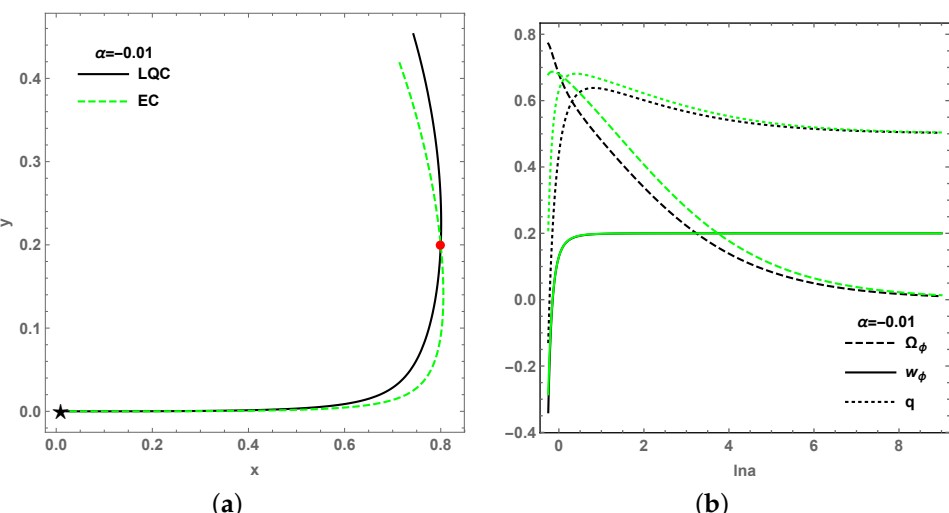

**Figure 13.** (**a**) The $x$-$y$ phase plane projected from 4-dim system of $\{x, y, z, \sigma\}$ for both LQC(black) and EC(green dash) around $P_{e2}$, corresponding to the parameters of $\gamma_k = \frac{6}{5}$, $\Gamma = \frac{6}{5}$ and $\alpha = -0.01$. The initial points are $(0.8, 0.2, 0.1, 4)$ for LQC, and $(0.8, 0.2, 0, 4)$ for EC. The star stands for the final state of the evolution, while the solid circle stands for the initial point. (**b**) the evolution of the cosmological quantities of $\Omega_\phi$, $w_\phi$ and $q$ for both LQC(black) and EC(green) cases around $P_{e2}$.

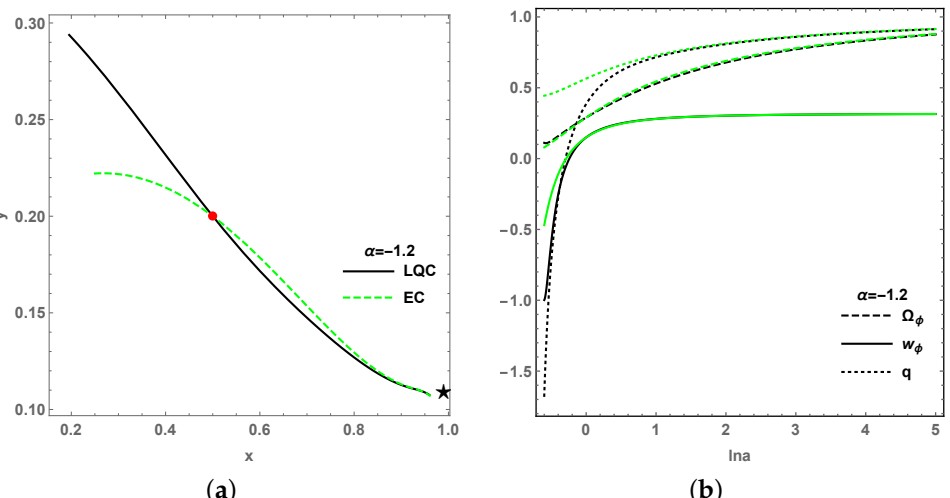

**Figure 14.** (**a**) The *x*-*y* phase plane projected from 3-dim system of $\{x,y,z\}$ for both LQC(black) and EC(green) around $P_g$, corresponding to the parameters of $\gamma_k = \frac{4}{3}$, $s = 4$ and $\alpha = -1.2$. The initial points are $(0.5, 0.2, 0.1)$ for LQC, and $(0.5, 0.2, 0)$ for EC. The star stands for the final state of the evolution, while the solid circle stands for the initial point. (**b**) the evolution of the cosmological quantities of $\Omega_\phi$, $w_\phi$ and $q$ for both LQC(black) and EC(green) cases for $P_g$.

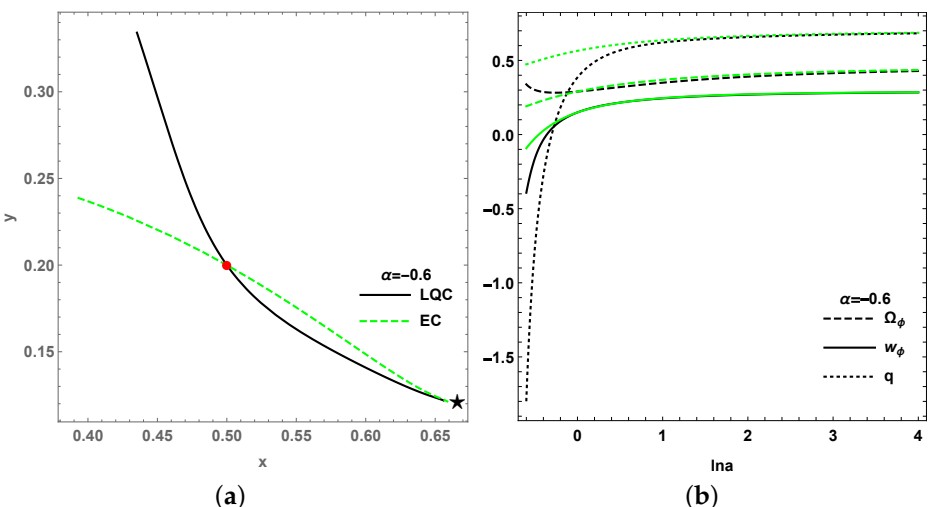

**Figure 15.** (**a**) The *x*-*y* phase plane projected from 3-dim system of $\{x,y,z\}$ for both LQC(black) and EC(green) around $P_h$, corresponding to the parameters of $\gamma_k = \frac{4}{3}$, $s = 4$ and $\alpha = -0.6$. The initial points are $(0.5, 0.2, 0.1)$ for LQC, and $(0.5, 0.2, 0)$ for EC. The star stands for the final state of the evolution, while the solid circle stands for the initial point. (**b**) the evolution of the cosmological quantities of $\Omega_\phi$, $w_\phi$ and $q$ for both LQC(black) and EC(green) cases for $P_h$.

Finally, we discuss the evolution of $\dot{H}$, $H$ and $\rho$, choosing certain parameter values and initial values in another dynamical system, formed as in Model I, as below:

$$\dot{\phi} = \dot{\phi}, \tag{73}$$

$$\ddot{\phi} = \frac{-6HA\eta X^\eta - Bn\phi^{n-1} - \alpha H(\rho - A(2\eta-1)X^\eta - B\phi^n)\dot{\phi}^{-1}}{A\eta(2\eta-1)X^{\eta-1}}, \tag{74}$$

$$\dot{H} = -\frac{1}{2}(\rho + AX^\eta - B\phi^n)(1 - 2\frac{\rho}{\rho_{lc}}), \tag{75}$$

$$\dot{\rho} = -3H(\rho + AX^\eta - B\phi^n). \tag{76}$$

With the effect of LQC on cosmological evolution, we can predict that the future universe will experience an oscillating processes, which avoidd the Big Rip singularity and curvature singularity as $H$ and $\dot{H}$ are bounded. Figure 16 is based on $P_{e2}$ for different initial values of $H_0$ and $\rho_0$, from which the bouncing period is prolonged by larger values of $H_0$ and $\rho_0$. Figure 17a is based on $P_{h0}$ for no interaction, i.e., $\alpha = 0$, while Figure 17b is based on $P_g$ for coupling parameter $\alpha = -1.2$, which shows that $\alpha$ has little effect on the bouncing. The initial values of $\phi_0$ and $\dot{\phi}_0$ based on definition (45).

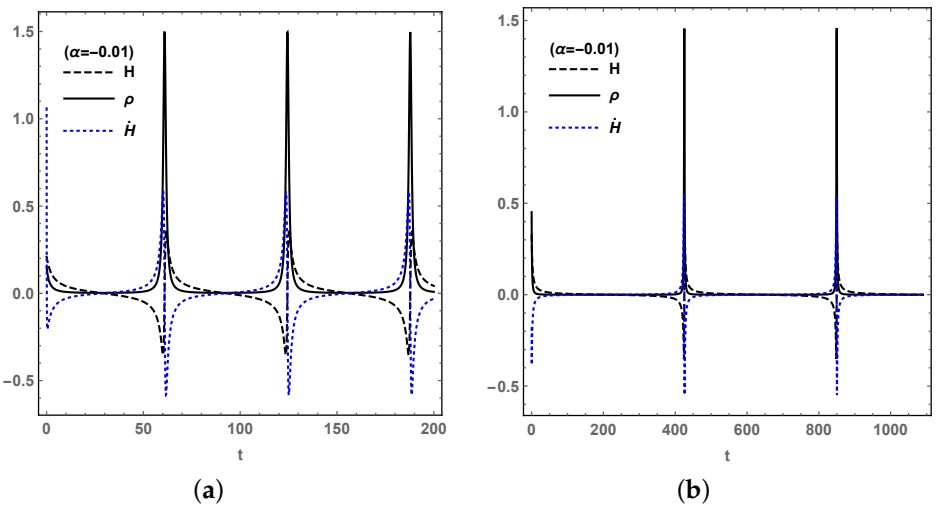

         (**a**)               (**b**)

**Figure 16.** For $P_{e2}$, the bouncing happens in LQC, with parameters $\eta = 3$, $n = -5$, $A = 0.002$, $B = 0.002$ and coupling parameter $\alpha = -0.01$. (**a**) The initial values $\phi_0 = 0.6$, $\dot{\phi}_0 = 1.94$, $H_0 = 0.21$ and $\rho_0 = 0.15$. (**b**) The initial values $\phi_0 = 0.5$, $\dot{\phi}_0 = 2.3$, $H_0 = 0.3333$ and $\rho_0 = 0.5$.

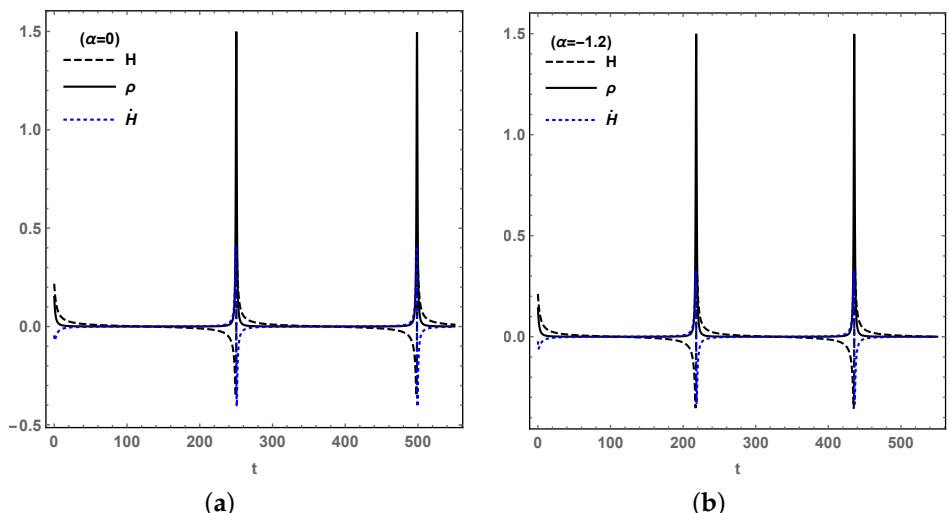

         (**a**)               (**b**)

**Figure 17.** The bouncing happens in LQC, with parameters $\eta = 2$, $n = -4$, $A = 0.5$, $B = 0.3$, with the initial values $\phi_0 = 1.85$, $\dot{\phi}_0 = 0.65$, $H_0 = 0.212$ and $\rho_0 = 0.15$. (**a**) $\alpha = 0$ for stable point $P_{h0}$; (**b**) $\alpha = -1.2$ for stable point $P_g$.

## 4. Conclusions

In summary, we have investigated the dynamical stability, cosmological evolution, and bouncing universe for the two kinds of k-essence DE models in depth, using the frame of LQC with the interaction $Q = \alpha H \rho_m$ in FRWL spacetime. We not only numerically analyzed the dynamical stabilities for Model I with $\mathcal{L}_1 = F(X)V(\phi)$ and Model II with $\mathcal{L}_2 = F(X) - V(\phi)$, respectively, and obtained the four stable points among the six critical points for Model I, the five stable points among the eleven critical points for Model II,

but also discussed the influences of coupling parameter and potential parameter on the evolutions of several cosmological quantities (such as the density parameter $\Omega_\phi$, EoS of DE $w_\phi$ and deceleration parameter $q$) around some stable points, especially the comparison between the cosmological evolutions in LQC with those in EC. The research results show that the EoS of DE $w_\phi$ is more sensitive to both coupling parameter and potential parameter than the deceleration parameter $q$ and the density parameter $\Omega_\phi$. Moreover, we found that the evolutions of $\Omega_\phi$, $w_\phi$ and $q$ in the last-time universe in LQC are in accordance with those in EC, although, in the early time, the differences from LQC to EC are great according to the quantum effects. It follows that the results given by us for the k-essence models in LQC could be viewed as a generalization of the results in EC. Finally, with the loop quantum gravity effects, we also achieved the bouncing universe in the two kinds of k-essence models for suitable initial values, which helps to avoid the future singularities and curvature singularity. Obviously, the bounce periods are evidently effected by the initial values, but the interaction does not effect the periods too much.

**Author Contributions:** Methodology, Y.W.; software, B.C., J.C., W.L. and Y.H.; supervision, Y.W.; writing—original draft, B.C.; writing—review and editing, Y.W. All authors have read and agreed to the published version of the manuscript.

**Funding:** This research was funded by the National Natural Science Foundation of China grant number 12075109, 12175096, 12175095, 11865012, and LiaoNing Revitalization Talents Program grant number XLYC2007047.

**Data Availability Statement:** Not applicable.

**Conflicts of Interest:** The authors declare no conflict of interest.

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
