# Peer review of "The Phase Space Analysis of Interacting K-Essence Dark Energy Models in Loop Quantum Cosmology"

_universe, doi:10.3390/universe8100520_

Round 1

Reviewer 1 Report

I send my comments in attach file

Author Response

Dear Reviewer,
    Thank you very much for the approval of our research about k-essence DE models in the frame of LQC.  Based on this work, we will keep on researching this kind of model in future. 
Best regards.

Reviewer 2 Report

The manuscript entitled as "The phase space analysis ..........loop quantum cosmology", which deals with two kinds of k-essence dark energy models in the frame work of LQC. The paper is good and well written. 

Some of my important criticism about the manuscript that I would like to point out are as follows:

1) The authors must take care of grammatical/spelling errors and subject verb agreement throughout the document.

2) In para-1 of Introduction, in last line, "DBI model" must be written in abbreviation. 

3) The motivation for taking the research problem is not given properly in Introduction. This needs latest references to be cited in the paper.

4) To improve the quality of Introduction the following references must be cited between [14-16] on page-1.

(i) Mod. Phys. Lett. A, Vol. 35, Issue 04, 10 Feb (2020) 2050002.

(ii) New Astronomy, Vol. 80, Issue October (2020) 101406. arXiv:1907.12968.

(iii) Chin. J. Phys. Vol. 73, Issue October (2021) 56-73.

(iv) Int. J. Geom. Method Mod. Phys. https://doi.org/10.1142/S0219887822501985.

5) To improve the quality of Introduction the following references must be cited after [37], the k-essence model on page-2 in para-2:

(i) New Astronomy, Vol. 68, Issue April (2019), 57-64.

(ii) Europ. Phys. J. Plus, Vol. 135, Issue 03 July (2020) 541.

(iii) New Astronomy, Vol. 83, Issue Feb (2021),  101478., arxiv:2011.07321.

6) To improve the quality of Introduction the following references must be cited between [52-57], on page-2 in para-3:

(i) arxiv:1906.00450, Pramana-J. Phys. Vol. 93 Issue6, Dec (2019) 89.

(ii) New Astronomy, Vol. 78, Issue 10 July (2020), 101368, arXiv:2002.11486.

7) In the manuscript, the authors have taken two parameters, the potential one "s" and the interaction one "alpha". These parameters have been considered different values in different figures of the paper. Why? Please give the explanation.  The author have chosen without any motivation.  

8) Whys not the two parameters "s" and "alpha" have been found by using the observational constraints otherwise it is only a mathematical study.

Hence, in this form, this would not be proper to publish it, if the revised version of the MS addresses al my comments, this paper could be suitable for publication in Universe. 

Author Response

Dear Reviewer 

  Thank you very much for reading our paper and providing the suggestions to improve our work. We have modified the grammatical/spelling errors, and cited the references to make the paper following the latest research. We have rewritten the motivation of different values of potential parameter and coupling parameter. Also, the observational constrains are considered, which makes the work more rational. The revised contents in the resubmitted paper are marked in red. For details, please see the attachment.

Thanks again for your help.

Best Regards

Round 2

Reviewer 2 Report

I have seen the revised manuscript, which is much improved now. The authors have incorporated all comments correctly in the revised MS. I recommend the paper for publication in Universe.